# Preharvest Mandarin Rind Disorder: Insights into Varietal Differences and Preharvest Treatments Effects on Postharvest Quality

**DOI:** 10.3390/plants13081040

**Published:** 2024-04-09

**Authors:** Alaaeldin Rezk, Tariq Pervaiz, Greg Douhan, David Obenland, Mary Lu Arpaia, Ashraf El-kereamy

**Affiliations:** 1Department of Botany and Plant Sciences, University of California Riverside, Riverside, CA 22963, USA; alaaeldin.rezk@ucr.edu (A.R.); tariq.pervaiz@ucr.edu (T.P.); mlarpaia@ucanr.edu (M.L.A.); 2Department of Microbiology & Plant Pathology, University of California Riverside, Riverside, CA 22963, USA; gregdo@ucr.edu; 3US Department of Agriculture (USDA), Agricultural Research Service, San Joaquin Valley Agricultural Sciences Center, Parlier, CA 93648, USA; david.obenland@usda.gov

**Keywords:** citrus rind disorder, Satsuma Owari mandarins, postharvest, non-chilling rind disorder

## Abstract

The citrus industry loses a significant amount of mandarin fruits either before or shortly after harvesting due to rind disorder. Different citrus cultivars are impacted by a physiological rind disorder that lowers fruit quality and marketability. Although the primary etiology of this condition is unknown, changes in relative humidity (RH) and rind water status can make it worse. The damage is initiated in the fall, especially following rain. It begins with irregular water-soaked areas that develop into dark-brown, necrotic lesions covering large portions of the fruit’s surface. The damage is evident in some citrus types such as Satsuma Owari mandarins and other cultivars. In this study, we attempted to understand and control the occurrence of this kind of rind disorder in Satsuma Owari mandarins growing under California conditions. Our data showed that fruit located in the outer part of the canopy suffer more than fruit in the interior canopy. We were able to reduce this damage in Satsuma Owari mandarins by applying 2,4-dichlorophenoxyacetic acid (2,4-D) at 16 milligrams/Liter (mg/L), gibberellic acid (GA_3_) at 20 mg/L, or Vapor Gard^®^ at 0.5 percent (*v*/*v*) at the color break stage. However, GA_3_ caused a delay in color development by approximately four weeks. GA_3_-treated fruit changed their color completely four weeks after the control, and the rind damage was at a very low percentage. Delaying rind senescence could be a good strategy to reduce the damage in mandarin orchards. Data showed that in addition to the benefits of the different treatments on preventing rind disorder at harvest, they have some beneficial effects during storage for four weeks either at 0.5 or 7.5 °C.

## 1. Introduction

The pre and postharvest quality of fruits relates to a wide range of factors, including market quality, appearance, nutritional quality, and more. Preharvest factors greatly contribute to fruit shape, color, texture, and taste [1]. Therefore, the postharvest shelf life of fruits is generally dependent on preharvest factors. Citrus fruits are prone to develop a variety of physiological rind disorders, which are present with a wide range of symptoms during handling and storage. Up to 60% of the overall output of fruit might be impacted by physiological disorders [2]. Physiological rind disorders include rind breakdown, chilling injury, non-chilling rind/peel pitting and rind staining, peteca spots, puffiness, creasing, stylar end breakdown, oleocellosis, and stem-end rind breakdown (SERB) [3,4,5]. Externally, non-chilling physiological rind disorder symptoms may seem similar to postharvest chilling injury symptoms. The majority of non-chilling rind disorders affect both the albedo and the flavedo tissues, and the fruit may be susceptible to the disorder while still connected to the tree [4,5].

The application of exogenous plant hormones such as auxins, gibberellins, and cytokinins has been investigated to determine their impact on the slowing of senescence-related changes in citrus fruits [6,7]. Auxins, a group of plant growth regulators (PGRs), have a direct impact on abscission by delaying fruit senescence, which enhances citrus fruit quality and yield. For reducing physiological disorders, naphthyl acetic acid, 2,4-dichlorophenoxyacetic acid, and gibberellic acid have been widely used [8,9]. Pre-harvest applications of gibberellic acid (GA_3_) on fruits have been tested to delay senescence, increase epicarp firmness, postpone harvest time, and control physiological disorders, all with contrasting outcomes depending on the species, variety, dose, and application technique, among other variables [6,10]. GA_3_ pre-harvest treatment has been shown to decrease fruit drops and puffiness while delaying softening and rind color. Mandarin fruits’ ripening and fruit degradation were both postponed by pre-harvest GA_3_ spraying. GA_3_ was the most successful in making fruits heavier and less acidic overall. It also helps to reduce chlorophyll loss in citrus fruit [11,12]. Exogenous plant growth regulators significantly reduced fruit drop and increased fruit weight, juice percentage, total soluble solids, acidity, and vitamin C content [11,13]. In addition, GA_3_ delayed maturity/senescence and decreased postharvest losses recorded in ‘Kinnow’ mandarin [14,15]. Mcdonald et al. [16] reported that GA_3_ treatments enhanced peel oil while having no impact on fruit quality. 2,4-dichlorophenoxyacetic acid has been used as a postharvest packhouse treatment to delay calyx abscission (to repress postharvest decay) or calyx senescence occurring as a result of ethylene degreening treatment, thereby maintaining the quality of citrus fruits [17,18,19]. Synthetic auxin 2,4-D is quickly absorbed, translocated, and deposited in the phloem of young meristematic tissue, where it accumulates in sink organs like young leaves, flowers, or fruitlets and promotes cell growth [20,21,22]. Furthermore, previous studies have shown that 2,4-D exhibits broad-spectrum antifungal action in vitro and has the potential to significantly lower the incidence of black rot in citrus fruits [23]. Grapefruit collected in October and November had greater levels of chlorophyll and beta-carotene than fruit gathered during the rest of the season. Preharvest GA_3_ treatments used before fruit color break promote improved resistance to the Caribbean fruit fly, whereas late-season grapefruit are more vulnerable to Caribbean fruit fly [24]. GA_3_ delayed fruit coloring and rind softening, while 2,4-D greatly decreased fruit drop [25]. In Australia and California, navel orange peel stiffness has been increased and color development has been delayed using GA_3_ sprays, usually in combination with 2,4-D. Additionally, GA_3_ with 2,4-D has been applied in Florida and South Africa to enhance peel stiffness and delay the development of off-color in grapefruit [26,27,28]. Plant growth regulators often have little effect on the internal quality of citrus fruits, such as the juice content, brix, and the acid ratio [29]. Additionally, they prolong storage life, maintain rind hardness, and lessen fruit weight loss.

According to recent reports, moisture seeping through the fruit’s surface caused a preharvest rind problem in mandarins in California [30]. The same disorder was also observed in the laboratory following the fruits being soaked in water. It seems that water causes oxidative stress on the fruit’s surface. Therefore, in the present study, we investigated the pre and postharvest control of mandarin rind disorder with the application of phytohormones (GA_3_ and 2,4-D), and Vapor Gard (anti-transpirant concentrate) on the pre-harvest rind damage and the effects of these treatments on the fruit quality during storage at two different temperatures. Vapor Gard is a water-emulsifiable organic compound and is used on plants to reduce water transpiration. The soft, flexible film that forms when the spray treatment dries will greatly limit the amount of moisture loss by plant leaves and fruits.

## 2. Results

### 2.1. Physical Assessment of Rind Disorder Symptoms

Data recorded from the two growing seasons revealed that the damage started in the fall, right before Satsuma ‘Owari’ mandarins were harvested, and was preceded by raindrops depositing water on the fruit rind. It began with irregular water-soaked areas (spots) that developed into dark-brown, necrotic lesions covering large portions of the fruit’s surface (Figure 1A–F). The same symptoms were induced in the laboratory by soaking the fruits of Satsuma ‘Owari’ mandarins in water for 24 h followed by drying at room temperature for another 48 h. The symptoms we obtained in the laboratory were comparable to those observed in the field (Figure 1G–I). The fruit damage was determined at the commercial harvest stage by counting the number of damaged fruits from outside and inside the canopy of the various mandarin varieties.

In the postharvest studies, the incidence of rind disorder was graded on a scale of 1 to 5 for the degree of damage, where 1 represents no damage, 2 = very slight damage, 3 = slight damage, 4 = moderate damage, and 5 = severe damage (Figure 2). To assess the extent of the damage, fruits were opportunistically inspected both on trees and in storage. On a tree, fruits were divided into the outer and inner canopy, and the data depicted that the pre-harvest rind disorder development was significantly higher in the outer canopy area of Satsuma Owari mandarins (Figure 3). The damage was absent in all other mandarin varieties including ‘Page’, ‘W. Murcott’, and ‘Tango’, hence we could not determine the efficacy of the treatments on controlling the preharvest rind disorder in treated fruits. The results of the Satsuma ‘Owari’ during the two consecutive seasons were as follows: the control treatments exhibited a significant number of damaged fruits in the outer canopy, followed by Vapor Gard and 2,4-D; however, the percentage of damaged fruits was substantially lower in the interior part of the canopy (Figure 3). In Satsuma mandarin, it appears that treatments with GA_3_, Vapor Gard, and 2,4-D at the color break stage have a major impact in lowering rind disorder. However, no significant difference was observed in the treated samples of the other three varieties (‘Page’, ‘W. Murcott’, and ‘Tango’) due to the absence of the rind disorder symptoms at the harvest stage. Additionally, the effect of these treatments on the other fruit quality parameters was determined. This information was collected to better understand the impacts of the treatments on fruit quality following storage.

### 2.2. Effect of Various Treatments on Fruit Quality at Harvest

Fruit color is one of the most important fruit qualities that determine marketability. In the current study, we determined the fruit color index by dividing the “a” by the “b” values obtained from the Minolta colorimeter; the value of the color index negatively correlated with the fruit orange color. Our data showed that the color break stage treatment with GA_3_ at 20 ppm reduced the color index at harvest, causing a delay in harvest time (Figure 4 and Figure 5). The effect was more pronounced in Satsuma, followed by ‘Tango’, than the ‘W. Marcotte’ and ‘Page’ during the two seasons. Vapor Gard and 2,4-D did not have a significant effect on fruit color during the two years of the study (Figure 4 and Figure 5).

Data showed that the GA_3_ treatment increased the fruit firmness during the first season (2019) compared to the control untreated fruit in ‘Owari’, and the same increasing trend was observed during the second season in ‘Owari’ and ‘Page’; however, this trend was less obvious in ‘W. Murcott’ and observed only in ‘Tango’ mandarin during the 2020 season (Figure 6). The effect of the various treatments on fruit weight and size showed no defined trend. A significant increase in fruit weight was observed following GA_3_ treatment in ‘Tango’ in 2019 and in ‘W. Murcott’ in 2020 only. It seems that none of the treatments showed a significant difference in the fruit weight, width, and length (Appendix A). In ‘Owari’, a significant reduction in brix was observed following GA_3_ treatment in the 2020 season only (Appendix A). In ‘Page’ mandarin, no significant differences were observed among treatments during both years relative to the control. In ‘W. Murcott’, all treatments significantly reduced the brix during the two seasons (Appendix A). In ‘Tango’, no significant effect of the treatments on the brix was observed. The response of the titratable acidity to the various treatments showed different patterns in the different varieties and from year to year (Appendix A). In 2019, GA_3_ treatment was the only treatment that significantly reduced the titratable acidity in ‘W. Murcott’ and ‘Tango’ mandarin. In 2020, a similar trend was observed, however, the reduction was only significant in ‘W. Murcott’, whereas 2,4-D and GA_3_ treatments increased the acidity in ‘Page’ mandarin in 2020 only.

The TSS/acid ratio determines the sweetness of the fruits. Our data showed that despite the lower brix values obtained following 2,4-D and GA_3_ treatments, the TSS/acid ratio did not show any significant reduction compared to the control except for an increase in ‘W. Murcott’ during 2019 and a reduction in ‘Page’ mandarin during the 2020 season (Appendix A).

Fruit juice pH showed no significant differences among treatments in ‘Owari’, ‘Page’, and ‘W. Murcott’ mandarin. However, Vapor Gard and GA_3_ significantly increased the juice pH in ‘Tango’ in the 2019 season only (Appendix A).

### 2.3. Postharvest Responses to Various Treatments Following Storage

After a four-week storage period, the final visual evaluation of the fruits revealed a significant difference in the severity of damage between the two storage temperatures (Table 1). The damage was notably higher at 0.5 °C compared to 7.5 °C during both years. At the lower temperature, the damage was significantly lower in fruits that received a pre-harvest treatment with GA_3_ and 2,4-D. At 7.5 °C, no significant difference in damage was observed with the GA_3_-treated fruits compared to the control; however, the 2,4-D treatment showed the next best results. It seems that Vapor Gard had little effect on the development of damage during the two seasons. Examining the different varieties, ‘Owari’ exhibited significantly higher damage under both temperature conditions (0.5 and 7.5 °C). Overall, within each variety, the pre-harvest GA_3_ treatment significantly reduced the extent of fruit damage. These findings strongly suggest that the application of GA_3_ as a pre-harvest treatment during the color break stage can significantly mitigate fruit damage during the subsequent four weeks of storage, regardless of the storage temperature being 0.5 °C or 7.5 °C. When examining various treatments, our data reveal that, for ‘Owari’ and ‘Tango’ varieties, the application of GA_3_ treatment led to an increase in fruit firmness at temperatures of 0.5 °C and 7.5 °C in both seasons (Table 2).

A more detailed analysis of individual varieties demonstrated that GA_3_ treatments consistently improved fruit firmness in ‘Owari’ and ‘Tango’ mandarins over the course of both years (Table 2). On the other hand, for ‘Page’ and ‘W. Murcott’ mandarin varieties, none of the treatments exhibited a statistically significant effect on fruit firmness, regardless of storage temperature or year (Table 2). The fruit color index value exhibits a negative correlation with the true orange color of the fruit, with lower values indicating greener fruits. Across all varieties, treatments, and storage, it was observed that storing mandarin fruit at a higher temperature (7.5 °C) contributed to increased development in the orange color, as evidenced by the higher color index (Appendix A). Examining each specific variety, the data revealed that the GA_3_ treatment consistently resulted in a lower color index compared to all other treatments, regardless of the season (Appendix A). The measurement of fruit sugar content using brix values suggested that storage temperature had no discernible impact on fruit sugar content, as there were no significant differences between fruits stored at 0.5 °C and 7.5 °C (Appendix A). However, when examining preharvest treatments, it was evident that fruits treated with GA_3_ displayed lower brix values in comparison to other treatments and controls. Nevertheless, this difference was statistically significant only in the case of Satsuma mandarins stored at both temperature settings throughout both seasons (Appendix A). Fruit titratable acidity was consistently higher at 7.5 °C compared to 0.5 °C over the course of the two study seasons (Appendix A). Aggregated across all varieties, the application of GA_3_ treatments led to a significant decrease in fruit titratable acidity during both years and at both temperature settings (Appendix A). Notably, the reduction in acidity following the GA_3_ treatments was more pronounced at 7.5 °C in ‘Owari’ mandarins, while at 0.5°C, the reduction was not statistically significant (Appendix A). In the case of ‘Page’ mandarins, no significant changes in titratable acidity were observed among different treatments and temperatures (Appendix A). On the other hand, the TA of ‘W. Murcott’ mandarins treated with GA were lower than controls in both 2019 and 2020 and at both storage temperatures.

Finally, in the ‘Tango’ variety, GA_3_ treatments reduced fruit acidity at both temperatures in 2019, but no significant difference was observed in 2020 at both temperatures (Appendix A). Fruit pH was generally higher in the fruits stored at 7.5 °C compared to those stored at 0.5 °C, as indicated in Appendix A. However, the difference was not large enough to be physiologically viable. Additionally, there were no significant differences found among treatments within each variety at both temperatures (Appendix A). The fruit TSS/acid ratio was higher in the fruits stored at 7.5 °C compared to those stored at 0.5 °C, as depicted in Appendix A. In the 2019 season, no significant differences were observed among treatments within each variety (Appendix A). However, in 2020, ‘Page’ mandarin exhibited a lower TSS/acid ratio following GA_3_ treatment at both temperatures. For ‘W. Murcott’, the GA_3_ treatment resulted in an increased TSS/acid ratio compared to other treatments, but this was observed only at 0.5 °C (Appendix A).

## 3. Discussion

### 3.1. Preharvest Rind Disorder

Mandarin preharvest rind disorder is a physiological condition that manifests after rain events that occur before harvest. This issue primarily impacts early varieties, such as Owari Satsuma, and carries economic consequences. The first documented occurrence of this damage was by Adaskaveg et al. [30] and Pervaiz et al. [5], and our data align with their findings. Our research indicates that this damage begins as clear areas on the rind that progressively turn brown over time, consistent with the observations made by Adaskaveg et al. [30]. This damage primarily results from the extended exposure of the fruit to standing water on the rind. We successfully replicated this condition in our laboratory using the methodology outlined by Adaskaveg et al. [30], as illustrated in Figure 1. Initially, we included four different Mandarin varieties in our study. However, the damage was only noticeable in Satsuma ‘Owari’ mandarins when they were observed in the field. The damage was associated with higher precipitation during December in the two years, just before the Satsuma Owari harvest (Appendix A). Our findings suggest that this damage is more pronounced in fruits located outside the canopy, as they are more vulnerable to rain exposure. Notably, citrus fruits within the canopy exhibit distinct characteristics, including a thicker wax layer, as reported by Ritenour and Dou [31] and Romero and Lafuente [32]. In addition, a comprehensive metabolome analysis has revealed a wide range of differences between fruits inside and outside the canopy [5,33]. The occurrence of damage exclusively in Owari Satsuma may be attributed to genetic variations between Satsuma and other varieties, as well as differences in maturation times. Owari Satsuma is one of California’s earliest varieties and is typically close to harvest at the time of rain during both seasons. It would be intriguing to explore the physiological distinctions among these four varieties in future studies.

### 3.2. Field Treatments to Control Preharvest Rind Disorder

The primary objective of this study was to find a practical solution for mitigating preharvest rind damage in Satsuma mandarin. We pursued this goal by evaluating various field treatments, including Vapor Gard, 2,4-D, and GA_3_, applied at the color break stage. Our data revealed that these treatments had a substantial impact in reducing preharvest rind damage in Satsuma mandarin. All three treatments demonstrated a significant reduction in preharvest rind disorder, particularly in fruits positioned outside the canopy. It is important to note that the mechanisms of action may differ among these treatments. Vapor Gard, a coating material that acts as a protective barrier, effectively isolates the rind from the adverse effects of rain events, thereby safeguarding the fruits from damage. This product has been previously documented by Serra et al. [33], and is widely used in the California citrus industry. The damage reduction observed with the application of 2,4-D at the color break stage can be attributed to its ability to delay oxidative stress and senescence in the rind. This delayed response enables the rind to become more resilient to adverse weather conditions, including oxidative stress induced by rain [31,32,33,34,35]. Similarly, the reduction in damage resulting from GA_3_ application likely stems from its capacity to enhance rind strength and alleviate oxidative stress [36,37]. GA_3_ is renowned for its powerful effects in improving rind health and reducing creasing in Navel oranges [26,38]. Our findings are consistent with the prior literature in this field, which underscores the effectiveness of these treatments in addressing preharvest rind damage, particularly in fruits located outside the canopy.

### 3.3. Effect of the Color Break Stage Treatments on Fruit Quality at Harvest

While the three treatments had some effect on reducing preharvest rind disorder, it is worth noting that a delay in fruit coloration was observed specifically after the application of GA_3_. However, there were no significant changes in the organoleptic characteristics of the fruit. The delay in coloration was quantified using a color index and was consistent across all varieties over the two years of our study. This phenomenon of delayed coloration following GA_3_ treatments is a well-documented occurrence in citrus production, as previously reported by Besada et al., [39]. The delay in coloration is likely due to the inhibition of chlorophyll degradation by GA_3_ [39,40]. Delaying the senescence of the peel and rind enhances fruit quality, increases citrus fruit’s shelf life, and increases its commercial value [35,41]. In addition, spraying GA_3_ in the late summer or fall on citrus fruits can increase peel integrity and postpone fruit senescence. However, these GA_3_ treatments cause rind color changes to be delayed, leading to fruits with green rinds [37]. It appears that GA_3_ reduces preharvest rind disorder by slowing down the senescence and maturation processes. Future studies are needed to determine the optimal timing and GA_3_ concentration for reducing preharvest rind damage without adversely affecting fruit coloration. On the other hand, other treatments, such as 2,4-D and Vapor Gard, did not significantly impact fruit coloration. Vapor Gard, being a protective chemical coating, shields the fruit from the damaging effects of rainwater on the rind without causing any noticeable physiological changes in the fruits. 2,4-D, known for its role as an auxin derivative, acts as a plant hormone promoter, enhancing overall fruit growth and reducing oxidative stress at low concentrations, as reported by Ma et al. [23]. Through the modification of epicuticular wax shape and lignin content in the peel, 2,4-D enhances stress-defense capabilities and delays the senescence of postharvest citrus fruits by boosting exogenous auxin, endogenous ABA, and SA concentrations while lowering ethylene production [23,24]. These treatments appear to be effective in reducing preharvest rind disorder without compromising the fruit’s coloration. Exogenous GA_3_ treatments at the color change stage of citrus fruits preserved the fruit’s peel quality and decreased mesocarp cracking. Furthermore, pre-harvest spraying with 2,4-D alone or in combination with GA_3_ is an efficient method for improving the peel quality of fruit held on trees, minimizing fruit decay and rind senescence, and reducing late-season fruit loss [34,35,36].

In a similar vein to fruit color, the application of GA_3_ resulted in an increase in firmness, especially in Owari and Tango, in most cases. Firmness is typically associated with the maturation and ripening process in citrus, as indicated by Muramatsu et al. [42]. This finding provides further evidence supporting the mode of action of GA_3_ in reducing preharvest rind disorder by delaying rind maturation. It is well-documented that during citrus ripening, various physiological changes occur in the rind, including chlorophyll degradation and a reduction in firmness [34,38,42]. However, none of the treatments caused significant changes in fruit weight or size. This observation can be attributed to the timing of the application, which was conducted at the color break stage. At this stage, fruits are already fully mature, and any increase in size from this point until harvest is typically minimal [34,36,42]. While the impact of GA_3_ on fruit color was evident, its effect on fruit organoleptic characteristics was minimal. There was a slight reduction in fruit brix, titratable acidity, TSS/acid ratio, and pH values in some varieties, and this was observed only in the second year of the study. This could be due to the variation in fruit quality from year to year, as indicated in Appendix A, and/or the timing of the application. It appears that the treatments used did not significantly affect the organoleptic characteristics of the fruit. Similar data have been reported in previous studies, as documented by Coggins et al. [43].

### 3.4. Effect of the Color Break Stage Treatments on Postharvest Fruit Quality

It appears that postharvest storage temperature plays a crucial role in the development of rind disorder during storage. Our data demonstrated that fruits stored at a lower temperature of 0.5 °C were more susceptible to rind disorder compared to those stored at 7.5 °C. This observation aligns with previously reported findings from Chalutz et al. [44]. Intriguingly, preharvest GA_3_ application seems to reduce the occurrence of rind disorder during postharvest storage. In contrast to the preharvest data, rind damage was observed in all varieties during postharvest storage, suggesting that the postharvest damage is different from the preharvest rind disorder we observed in the field in the case of Owari only. Vapor Gard, on the other hand, appeared to not affect postharvest rind disorder. In contrast, both 2,4-D and GA_3_ significantly reduced the incidence of postharvest rind damage. This finding is consistent with the notion that Vapor Gard primarily serves as a protective barrier and may have limited effectiveness during the postharvest period. In contrast, 2,4-D and GA_3_ are plant hormones that influence physiological processes, delaying rind senescence, and their effects extend into the postharvest storage period. Similar results were previously reported by Chapman [34], who found that GA_3_ and 2,4-D treatments enhanced the postharvest storability of fruits. El-Otmani et al. [25] reported that GA3 delayed the coloring of fruit and the weakening of the peel. Fruit coloring did not change by 2,4-D alone; however, it did considerably lessen fruit drop and somewhat delay rind softening. During postharvest storage, fruit firmness was notably reduced in all varieties under all treatments and temperatures. However, GA_3_ application seems to improve fruit firmness during postharvest storage, whereas the effects of 2,4-D and Vapor Gard on firmness were not as pronounced. These results suggest that GA_3_ might delay rind senescence to maintain higher firmness levels, as supported by findings reported by Chapman [34].

Color development during postharvest storage in citrus fruits has been previously investigated and reported by Ge et al. [45]. Storage temperature significantly influences the development of orange color in citrus fruits after harvest. It appears that higher temperatures, such as 7.5 °C, enhance color development compared to lower temperatures, like 0.5 °C. This could be attributed to the fact that at higher temperatures, chlorophyll degradation is accelerated, allowing the orange color to become more pronounced [32,40,42,45,46]. However, it is worth noting that GA_3_-treated fruits still exhibited a lower color index compared to the other treatments and the control. Applications of GA_3_ have been shown to delay color change and peel senescence. Since fruits treated with GA_3_ frequently remain green, the application of GA_3_ needs to be carefully studied. For fruit color development, two to four months should usually elapse between the previous GA_3_ application and the planned harvest [37]. This may be due to the extended impact of GA_3_ on the rind’s physiology, as discussed by Singh et al. [47]. Similar to the preharvest data, fruit brix did not vary significantly among treatments. However, titratable acidity, pH, and the TSS/acid ratio increased at the higher temperature of 7.5 °C, indicating a potential alteration in the maturation process, perhaps due to the warmer temperature. This finding aligns with Tietel et al.’s [48] report of increased fruit color and maturation indices when citrus fruits were stored at 8 °C for four weeks. Porat et al. [40] reported that fruit color stability significantly depends on storage temperature; at 2 °C, fruit could remain green for up to five weeks, but at 6, 12, and 20 °C, the rate of de-greening increased gradually. GA_3_ applications successfully preserved the green color of the fruits, whether they were used as postharvest dip treatments or preharvest sprays.

## 4. Materials and Methods

### 4.1. Experimental Site and Plant Material

The present study was conducted at a commercial citrus orchard located in San Joaquin Valley, California, USA. The 10-year-old fruit-bearing Satsuma ‘Owari’ mandarin (*Citrus unshiu*), ‘Page’ (*Citrus reticulata*), ‘W. Murcott’ (*C. reticulata* × *sinensis*) and ‘Tango’ (*Citrus reticulata*) trees grafted on Carrizo rootstock were used. Complete randomized block design (CRBD) was used with the three treatments including 2,4-D, GA_3,_ and Vapor Gard, and each treatment was replicated four times. Reproduction of the fruit disorder in the laboratory was carried out using the protocol previously described [30]. Briefly, mature Satsuma mandarin fruit (cv. Owari) were submerged and incubated for 24 h in distilled water at a temperature of 25 °C. Subsequently, the fruits were left to dry at room temperature (25 °C) and monitored for the development of the disorder.

### 4.2. Pre-Harvest Application of Plant Growth Regulators

Two different phytohormones were used: 2,4-dichlorophenoxyacetic acid (2,4-D) at 16 mg/L in the form of CitriFix (Amvac Chemical Corporation, New Port Beach, CA, USA) and gibberellic acid (GA_3_) at 20 mg/L in the form of Falgro^®^ (Fine Americans Inc., Walnut Creek, CA, USA), and Vapor Gard at 0.5 percent (*v*/*v*) (Miller Chemical and Fertilizer LLC, Hanover, PA, USA). The treatments were applied once at the color break stage in mid-October during the 2019 and 2020 seasons. The spray volume per tree was adjusted based on the recommended 1875 L per hectare. The fruits were harvested during the commercial harvest date based on the control of untreated trees during the first week of December. A sample of 20 fruits from each replicate was used to carry out a physiological and physiochemical analysis, while another batch of 200 fruits from each replicate was harvested and sent to a semi-commercial pack line at Lindcove Research and Extension Center to be washed and waxed using Fruit-A-Peel^Tm^0 Material# 301911 (Fruit Growers, CA, USA). Fruits were stored at two different temperatures (0.5 °C and 7.5 °C) with 85% relative humidity (RH) at the cold storage facilities of the Kearney Agricultural Research and Extension Center for four weeks and then for one week at 20 °C (85% RH). Fruits were evaluated after the first and fourth weeks of cold storage.

### 4.3. Measurements of Physiochemical Characteristics

The fruit quality parameters were determined using 20 fruits from each replicate with a total of eighty fruits per treatment. Fruit quality including fruit weight, size, color, and firmness were recorded.

### 4.4. Determination of Fruit Firmness

The Agrosta (Agrosta^®^Firmtech 2020, Serqueux, France) device was used to determine fruit firmness. Firmtech gently presses the fruit surface to determine fruit firmness. When fruit is squeezed by the instrument load cell the force increases. The rate at which the force (grams) increases per unit of deformation (mm) is defined as firmness (grams/mm).

### 4.5. Measuring Fruit Color

Fruit color was measured externally at the equator on opposite sides of intact fruit. A colorimeter (Chroma Meter Cr-400 Konica Minolta Sensing, Inc., Tokyo, Japan) was used for measuring skin as L*, a*, and b* values. The fruit color index was calculated by dividing a* value by b*.

### 4.6. Fruit Juice Extraction

The chemical analysis was performed at harvest and after storage. Ten fruits from each replicate were randomly selected, and a hydraulic juice extraction machine was used to extract juice for physiochemical analysis.

### 4.7. Total Soluble Solids (TSS)

The TSS (brix◦) was measured by placing a few drops of juice on the prism of an ultraprecision digital refractometer (RFM110 Refractometer Bellingham Stanley Ltd. Tunbridge Wells, UK)

### 4.8. Titratable Acidity (TA) and pH

For TA quantification, 5 mL of clear juice samples were placed in a plastic beaker loaded on a computer-controlled automatic titrator Excellence T5 (Mettler-Toledo, OH, USA).

### 4.9. Experimental Design and Statistical Analysis

The field experiment was carried out in a complete randomized block design (CRBD). For laboratory experiments, a completely randomized design (CRD) was used. The statistical analyses were performed using R Statistical Software (v4.1.2; R Core Team 2021). The data were analyzed by two-way analysis of variance (ANOVA). Mean separations were performed by the least significant difference (LSD) method. Differences at *p* = 0.05 were considered significant.

## 5. Conclusions

In summary, our data suggest that Vapor Gard, 2,4-D, and GA_3_ treatments applied at the color break stage effectively reduce preharvest rind disorder in Satsuma Owari mandarin. However, the extended effect in reducing postharvest damage was observed only in fruits treated with 2,4-D and GA_3_. Further studies are needed to better understand the physiological differences among varieties and their responses to GA_3_ treatments.

## Figures and Tables

**Figure 1 plants-13-01040-f001:**
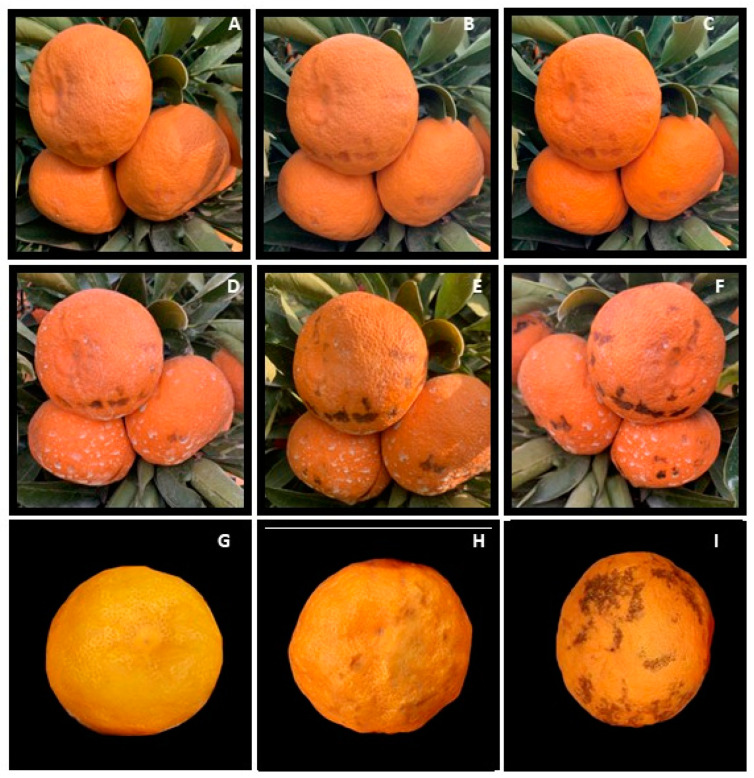
Initiation and development of the mandarin rind disorder following the rain event in the field (**A**–**F**) and the water soaking in the laboratory for 24 h followed by drying at room temperature for 0 time (**G**), 24 h (**H**) and 48 h (**I**).

**Figure 2 plants-13-01040-f002:**
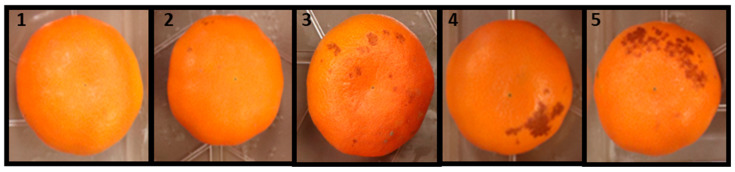
Rind disorder rating scale used in the study: 1 = no damage, 2 = very slight damage, 3 = slight damage, 4 = moderate damage, 5 = severe damage.

**Figure 3 plants-13-01040-f003:**
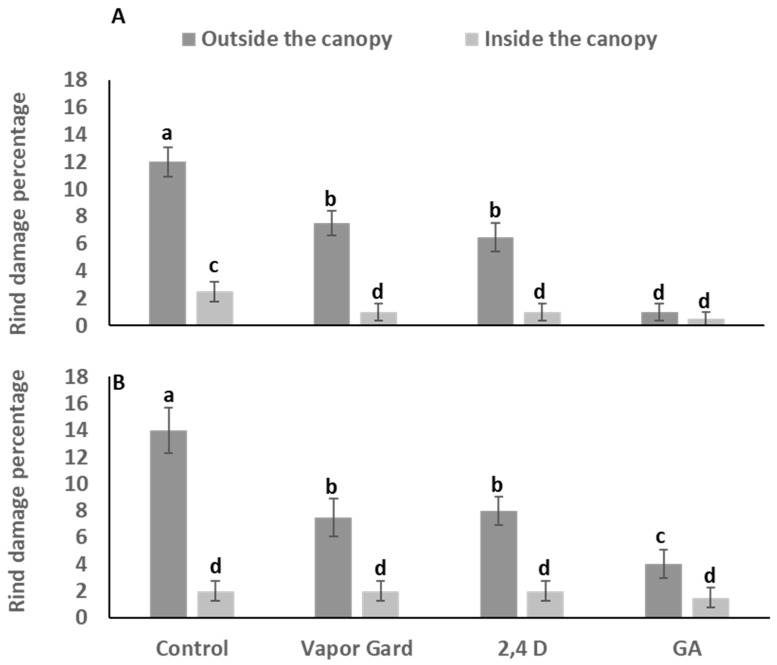
Effect of various treatments on the occurrence of the mandarin rind disorder in Satsuma ‘Owari’ mandarin fruits located either inside or outside the canopy during the harvest of 2019 (**A**) and 2020 (**B**). Different letters on bars indicate significant differences at *p* < 0.05 according to the Tukey HSD test within the same season. The error bars represents the standard error.

**Figure 4 plants-13-01040-f004:**
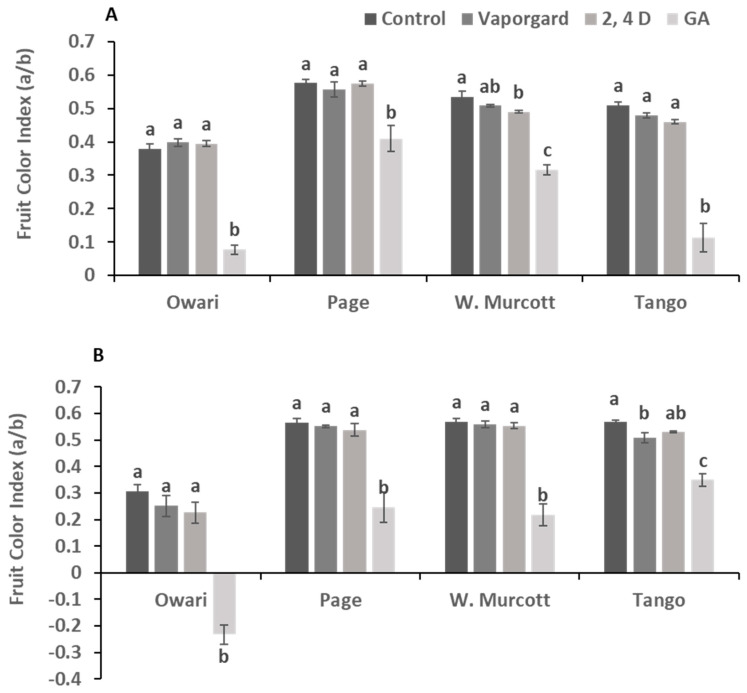
Effect of various treatments on the fruit color index of four mandarin varieties at harvest during the 2019 (**A**) and 2020 seasons (**B**). Letters indicate the difference among all treatments and varieties (*p* ≤ 0.05).

**Figure 5 plants-13-01040-f005:**
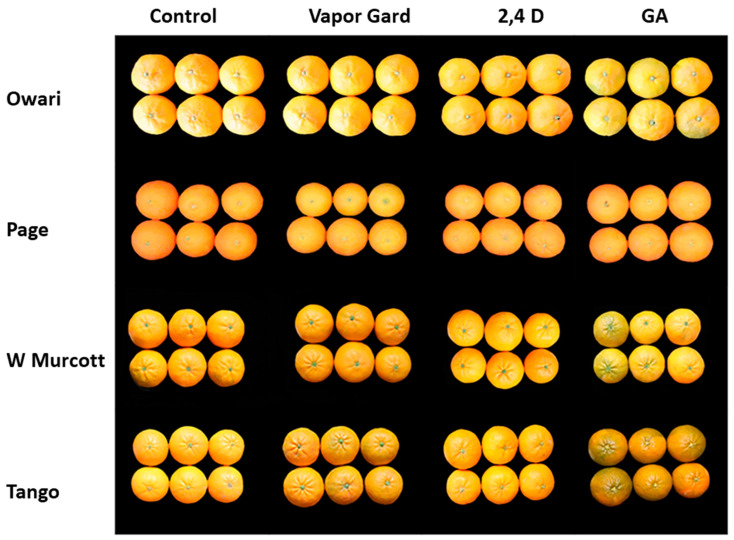
Fruit color of mandarin varieties in response to preharvest treatments at the harvest of 2019 season.

**Figure 6 plants-13-01040-f006:**
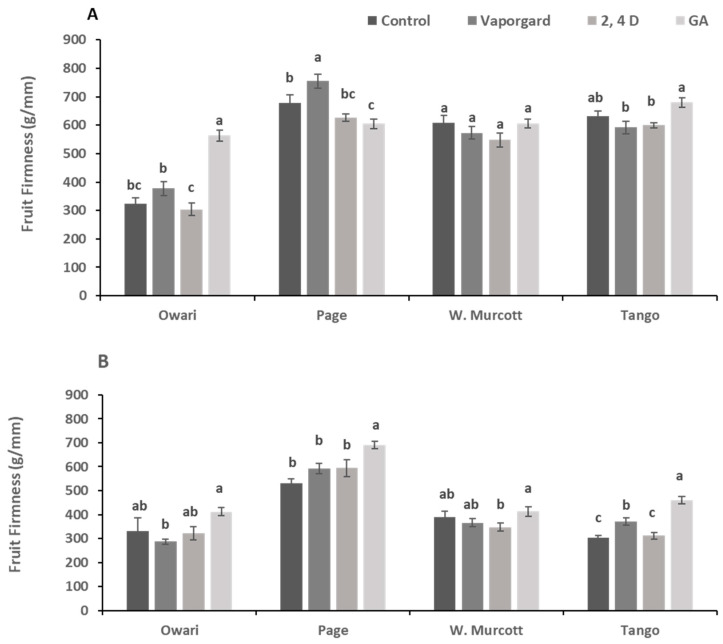
Effect of various treatments on fruit firmness of four mandarin varieties at harvest during the 2019 (**A**) and 2020 seasons (**B**). Letters indicate the difference among all treatments and varieties (*p* ≤ 0.05).

**Table 1 plants-13-01040-t001:** Rind disorder occurrence in four mandarin varieties during postharvest storage in response to various treatments in the 2019 (**1a**) and 2020 (**1b**) seasons. Fruits were stored at 0.5 °C and 7.5 °C for four weeks and then for one week at 20 °C. The number indicates the severity of rind disorder, i.e., 1 = no damage, 2 = 10% of the rind was damaged, 2 = very slight damage, 3 = slight damage, 4 = moderate damage, 5 = severe damage. Data are the average of four replicates, and each contains 20 fruits, ± the standard error; letters indicate the statistical difference among treatments and varieties at the level of 0.05.

1a											
	0.5 °C					7.5 °C					
	Control	Vapor Gard	2,4-D	GA	Mean	Control	Vapor Gard	2,4-D	GA	Mean	
**Owari**	3.9 ± 0.07 a	3.74 ± 0.1 ab	3.92 ± 0.08 a	3.45 ± 0.16 b	3.75 ± 0.07 a	3.04 ± 0.11 a	2.98 ± 0.29 a	2.79 ± 0.09 a	2.16 ± 0.15 b	2.74 ± 0.12 c	3.25 ± 0.11 B
**Page**	3.55 ± 0.27 a	3.14 ± 0.3 ab	2.68 ± 0.31 b	2.45 ± 0.22 b	2.95 ± 0.17 b	2.4 ± 0.12 a	2.22 ± 0.25 a	2.4 ± 0.08 a	2.28 ± 0.06 a	2.33 ± 0.067 d	2.64 ± 0.1 C
**W. Murcott**	3.6 ± 0.06 a	3.62 ± 0.14 a	3.7 ± 0.08 a	3.08 ± 0.13 b	3.5 ± 0.08 a	3.19 ± 0.19 a	2.88 ± 0.28 a	3.22 ± 0.06 a	2.68 ± 0.12 a	2.99 ± 0.10 b	3.245 ± 0.08 B
**Tango**	3.91 ± 0.17 a	3.47 ± 0.23 a	3.61 ± 0.14 a	3.75 ± 0.07 a	3.69 ± 0.08 a	3.52 ± 0.14 a	3.24 ± 0.07 a	3.2 ± 0.08 a	3.31 ± 0.18 a	3.31 ± 0.07 a	3.5 ± 0.06 A
	3.74 ± 0.09 a	3.55 ± 0.1 ab	3.42 ± 0.15 bc	3.18 ± 0.14 c		3.04 ± 0.12 a	2.95 ± 0.11 ab	2.78 ± 0.13 bc	2.61 ± 0.13 c		
3.47 ± 0.07 A						2.84 ± 0.06 B					
**1b**											
	**0.5 °C**					**7.5 °C**					
	**Control**	**Vapor Gard**	**2,4-D**	**GA**	**Mean**	**Control**	**Vapor Gard**	**2,4-D**	**GA**	**Mean**	
**Owari**	3.1 ± 0.06 a	3.22 ± 0.06 a	2.4 ± 0.08 b	3.09 ± 0.17 a	3.01 ± 0.11 a	2.52 ± 0.06 a	2.43 ± 0.08 a	1.99 ± 0.16 b	2.58 ± 0.03 a	2.38 ± 0.07 b	2.66 ± 0.08 B
**Page**	3.43 ± 0.06 a	3.05 ± 0.13 b	2.40 ± 0.12 c	3.17 ± 0.10 ab	2.95 ± 0.1 a	3.45 ± 0.21 a	3.37 ± 0.16 a	2.88 ± 0.11 b	3.15 ± 0.1 ab	3.21 ± 0.09 a	3.11 ± 0.07 A
**W. Murcott**	2.17 ± 0.13 b	2.04 ± 0.06 bc	1.74 ± 0.06 c	2.77 ± 0.16 a	2.18 ± 0.11 b	1.72 ± 0.14 a	1.66 ± 0.08 ab	1.44 ± 0.06 b	1.75 ± 0.07 a	1.64 ± 0.05 c	1.91 ± 0.08 C
**Tango**	2.23 ± 0.06 a	1.33 ± 0.1 c	1.4 ± 0.07 c	1.66 ± 0.05 b	1.65 ± 0.1 c	1.59 ± 0.03 a	1.19 ± 0.06 bc	1.28 ± 0.03 c	1.4 ± 0.06 b	1.36 ± 0.04 d	1.51 ± 0.06 D
	2.73 ± 0.14 a	2.43 ± 0.20 b	1.97 ± 0.13 c	2.67 ± 0.17 a		2.32 ± 0.20 a	2.19 ± 0.21 a	1.87 ± 0.17 b	2.22 ± 0.18 a		
2.45 ± 0.09 A						2.15 ± 0.1 B					

**Table 2 plants-13-01040-t002:** Fruit firmness (g/mm) of four mandarin varieties during postharvest storage in response to various treatments in the 2019 (**2a**) and 2020 (**2b**) seasons. Fruits were stored at 0.5 °C and 7.5 °C for four weeks and then for one week at 20 °C. Data are the average of four replicates, and each contains 20 fruits, ± the standard error; letters indicate the statistical difference among treatments and varieties at the level of 0.05.

2a											
	0.5 °C					7.5 °C					
	Control	Vapor Gard	2,4-D	GA	Mean	Control	Vapor Gard	2,4-D	GA	Mean	
**Owari**	140.45 ± 3.5 ab	136.79 ± 4.7 b	130.39 ± 2.5 b	159.70 ± 11.4 a	141.83 ± 4.1 d	149.1 ± 2.35 ab	157.52 ± 7.09 a	135.96 ± 3.43 b	160.05 ± 2.68 a	150.65 ± 3.1 c	146.24 ± 2.63 D
**Page**	333.91 ± 9.7 a	350.39 ± 21.7 a	351.77 ± 12.0 a	344.73 ± 15.1 a	345.2 ± 7.1 a	303.76 ± 10.74 a	317.2 ± 18.11 a	326.2 ± 24.45 a	306.23 ± 11.7 a	313.35 ± 8.02 a	329.27 ± 5.99 A
**W. Murcott**	199.93 ± 6.4 ab	202.04 ± 6.9 a	177.29 ± 4.6 b	191.9 ± 10.5 ab	192.79 ± 4.2 c	215 ± 6.81 a	206.73 ± 5.14 a	203.33 ± 19.08 a	235.88 ± 7.95 a	215.23 ± 5.97 b	204.01 ± 4.11 C
**Tango**	239.78 ± 9.0 ab	218.09 ± 6.7 b	245.2 ± 11.6a b	254.03 ± 8.3 a	239.27 ± 5.3 b	210.82 ± 7.25 b	204.21 ± 7.52 b	213.8 ± 7.51 b	276.1 ± 19.31 a	226.23 ± 9.11 b	232.75 ± 5.32 B
	288.52 ± 18.4 a	255.23 ± 21.1 a	277.75 ± 21.4 a	377.59 ± 18.9 a		219.7 ± 14.62 b	216.0 ± 17.39 b	225.22 ± 17.6b	244.56 ± 15.14a		
229.77 ± 9.8 A						226.36 ± 8.05 A					
**2b**											
	**0.5 °C**					**7.5 °C**					
	**Control**	**Vapor Gard**	**2,4-D**	**GA**	**Mean**	**Control**	**Vapor Gard**	**2,4-D**	**GA**	**Mean**	
**Owari**	112.84 ± 3.32 b	113.1 ± 7.39 b	132.96 ± 10.6 b	162.03 ± 8.07 a	128.11 ± 6.14 c	110.35 ± 1.58 b	109.09 ± 0.85 b	122.63 ± 3.1 a	121.94 ± 3.42 a	116 ± 1.97 d	118.05 ± 4.96 C
**Page**	228.2 ± 6.76 a	246.45 ± 4.56 a	248.21 ± 8.86 a	250.41 ± 9.72 a	243.32 ± 4.14 a	191.18 ± 1.32 b	214.54 ± 6.62 a	210.5 ± 5.84 a	206.3 ± 6.23 ab	205.63 ± 3.34 c	224.47 ± 4.28 B
**W. Murcott**	259.2 ± 7.04 a	225.85 ± 4.86 b	232.13 ± 3.9 b	258.85 ± 8.07 a	244.01 ± 4.8 a	247.1 ± 5.62 ab	239.8 ± 6.48 ab	226.6 ± 10.48 b	258.1 ± 1.96 a	242.9 ± 4.26 b	243.45 ± 3.16 A
**Tango**	194.96 ± 1.45 c	217.91 ± 9.28 b	204.8 ± 6.3 bc	247.06 ± 3.72 a	216.18 ± 5.72 b	238.11 ± 4.84 b	243.13 ± 7.35 b	235.03 ± 7.03 b	291.44 ± 10.52 a	251.93 ± 6.87 a	234.05 ± 5.44 A
	198.8 ± 14.28 b	205.8 ± 11.72 b	199.56 ± 13.5 b	234.09 ± 10.29 a		196.68 ± 14.1b c	205.03 ± 12.9 b	195.29 ± 13.42 c	219.45 ± 16.76 a		
205.9 ± 7.15 A						204.11 ± 7.12 A					

## Data Availability

All obtained data are included in this manuscript.

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
