# Peer review of "Preharvest Mandarin Rind Disorder: Insights into Varietal Differences and Preharvest Treatments Effects on Postharvest Quality"

_plants, 2024, doi:10.3390/plants13081040_

Round 1
Reviewer 1 Report
Comments and Suggestions for Authors
Editing of the manuscript is a must - the last paragraph of the Discussion is from Author Instructions, for example!
Authors do not state what statistical analyses were done to determine mean separations, and some of the ANOVA results for significant differences need to be critically analyzed (means +/- SEs not different for some treatments, and probably physiologically meaningless).
Figure 3: No mean separation indicated. Does the P < 0.05 indicate that there was a significant difference even in 2019 with the GA treatment between outer and inner canopy fruit? The legend should state that these data were collected at time of harvest (fruit not stored).
lines 145-146: Statement not supported by data - GA lowered fruit firmness of Page in 2019, but GA treatment of Page in 2020 not different from control
In line 159, authors start referring to 'W. Murcott' as WM without labeling it as such
line 165: GA increased TSS/Acid ratio in WM in 2019
line 169: VaporGard and GA increased, not decreased, juice pH in Tango in 2019
Table 1: Table not labeled (A) and (B) or 2019 and 2020. Add "storage" after "postharvest" in caption. Not clear what blue versus red mean +/- SE.
line 177: GA did not decrease injury compared to control in 2020 for fruit stored at 0.5C and at 7.5 C
line 180 & 181: Damage to Owari compared to other varieties differed by year and temperature (comparing blue means and sds); ANOVA table would be good addition to support authors' statements
line 187: No difference in fruit firmness in Owari and Tango treated with GA compared to control and stored at 0.5C in 2019; or for Owari stored at 7.5C in 2019
line 190: Are the authors referring to 'Owari' when they wrote "Satsuma"? Only difference in fruit firmness in Owari with GA treatment was for fruit in 2020; no difference in 2019
In line 192, do the authors mean "none" instead of "some"?
Authors should recheck statistical mean separaration. In Table S-1, for 2020 Tango fruit, mean color index was 0.53 +/1 0.01 at 0.5C and 0.52 +/1 0.01 at 7.5C, which do not seem statistically different or physiologically meaningful
line 198: GA did not change Color Index of Tango fruit stored at 0.5C compared o conrol in 2020 (Table S-1)
line 208: "Table 5" should be "Table S-3" , then delete "Table S-5" and "Table S-#" in lines 210 and 212; authors mean; authors mean "aggregated across all varieties", not "Regardless of fruit variety" because GA did not decrease TA in 2019 of Owari and Page fruit stored at 0.5C, and of Owari, Page, and Tango stored at 0.5C in 2020, and of Page fruit stored at 7.5 C in 2019, or of Tango fruit stored at 7.5C in 2020
line 214: suggest rewording to "TA of 'W. Murcott' mandarins treated with GA was lower than controls in both 2019 and 2020 and at both storage temperatures."
Table 2 legend should provide units for firmness. In 2020, was firmness of WM fruit REALLY significantl different if stored at 0.5C vs 7.5C (means 244 vs 243 g/mm)?
line 220: no sig diff between control and GA treated Tango stored at either temperature in 2020
line 222: there is no Supplementary Table 6; aggregated across varieties, GA treatments resulted in lower pH compared to the other treatments; in 2020 pH was greater for WM
treated with GA compared to the other treatments and in 2019 for Owari fruit stored at 7.5C, although it is debatable how physiologically significant those small differences were
line 224: there is no Table 7
line 227: no difference between control TSS/Acid ratio in 2020 for Page stored at 7.5C; ratio also different for WM and Tango treated with GA in 2020 and stored at 7.5C
line 306 statement not supported by 2019 data
line 312 statemnt contradicts statement in lines 208-210
line 331 should be qualifed by addition "in some years and for some varieties"
line 340: statement on fruit firmness not supported by treatment means +/- se compared to controls in Table 2
Extra period in line 378
Comments on the Quality of English Language
Author Response
We would like to thank the reviewer for the constructive comments. We have addressed all the comments in the attached file. Please see the attachment.

Reviewer 2 Report
Comments and Suggestions for Authors
This paper by Rezk et al report on a physiological disorder previously found in Owari mandarins. The disorder has been characterized previously under California conditions by researchers in the same research group and the etiology has been already well established by them. The present work focuses mostly on practical ways to alleviate the disorder.
I have some questions regarding methodology that do not me allow to accept this paper as presented:
In Material and Methods,
- please state clearly the treatments. How many replicates were used per treatment?
-Why was a positive control not included? If the authors mention that rainfall is involved, I expect to have some trees treated with water. Also, in this sense, how was the rainfall in both harvesting seasons during the period of time that the treatments were applied? this information is critical to understand the effectiveness of the treatments.
Some attention to detail in the writing is lacking:
For instance, there are ideas and sentences repeated: please compare lines 114-117 with lines 122-124. Also, lines 86-88 and 91-93.
Lines 368-371 are a strange addition that seem more a reviewer's comment than an author's claim.
Also, some discussion of oxidative stress is included but oxidative stress was not measured in this study, so we do not know how the treatments reduced oxidative stress in the conditions assayed.
Finally, the authors claim that the disorder they find in the lab with their system are the same than those during postharvest. I have 2 questions there: do the authors have any prove of this besides visual assessment (for instance, microscopy documentation of the damage? And second, the laboratory induction of the disorder is not described in the M&M section.
Comments on the Quality of English Language
Although in general is well written, there are some inconsistencies through the text:
Line 83, that sentence needs some revision.
W. Murcott is sometimes WM, others W Murcott. Please be consistent. Also Brix sometimes is with the B capitalized, others is brix. Same with 2,4-D as compared to 2,4 D. All these inconsistencies are found in many places in the text.
PH needs to be changed to pH through the manuscript.
Author Response

(The authors gave the same response as above.)
